# Superfluid motion and drag-force cancellation in a fluid of light

Claire Michel [1], Omar Boughdad[1], Mathias Albert[1], Pierre-Élie Larré[2,3] & Matthieu Bellec [1]

Quantum fluids of light merge many-body physics and nonlinear optics, revealing quantum hydrodynamic features of light when it propagates in nonlinear media. One of the most outstanding evidence of light behaving as an interacting fluid is its ability to carry itself as a superfluid. Here, we report a direct experimental detection of the transition to superfluidity in the flow of a fluid of light past an obstacle in a bulk nonlinear crystal. In this cavityless all-optical system, we extract a direct optical analog of the drag force exerted by the fluid of light and measure the associated displacement of the obstacle. Both quantities drop to zero in the superfluid regime characterized by a suppression of long-range radiation from the obstacle. The experimental capability to shape both the flow and the potential landscape paves the way for simulation of quantum transport in complex systems.

[1] Institut de Physique de Nice, Université Côte d'Azur, CNRS, Nice, France. [2] Laboratoire de Physique Théorique et Modélisation, Université de Cergy-Pontoise, CNRS, 2 Avenue Adolphe-Chauvin, 95302 Cergy-Pontoise CEDEX, France. [3] Laboratoire Kastler Brossel, Sorbonne Université, CNRS, ENS-Université PSL, Collège de France, 4 Place Jussieu, 75252, Paris CEDEX 05, France. Correspondence and requests for materials should be addressed to C.M. (email: claire.michel@unice.fr) or to M.B. (email: bellec@unice.fr)

Superfluidity was originally discovered in 1938[1] when a $^4$He fluid cooled below a critical temperature flowed in a non-classical way along a capillary[2]. This was the trigger for the development of many experiments genuinely realized with quantum matter, as with $^3$He fluids[3] or ultracold atomic vapors[4,5]. The superfluid behavior of mixed light-matter cavity gases of exciton-polaritons was also extensively studied[6,7], leading to the emergent field of "quantum fluids of light"[8]. Before being theoretically developed for cavity lasers[9,10], the idea of a superfluid motion of light originates from pioneering studies in cavityless all-optical configurations[11] in which the hydrodynamic nucleation of quantized vortices past an obstacle when a laser beam propagates in a bulk nonlinear medium was investigated[12]. In such a cavityless geometry, the paraxial propagation of a monochromatic optical field in a nonlinear medium may be mapped onto a two-dimensional Gross-Pitaevskii-type evolution of a quantum fluid of interacting photons in the plane transverse to the propagation[4]. The intensity, the gradient of the phase and the propagation constant of the optical field assume respectively the roles of the density, the velocity, and the mass of the quantum fluid. The photon–photon interactions are mediated by the optical nonlinearity. It took almost twenty years for this idea to spring up again[13–16], driven by the emergence of advanced laser-beam-shaping technologies allowing to precisely tailor both the shape of the flow and the potential landscape.

The ways of tracking light superfluidity are manifold. Recently, superfluid hydrodynamics of a fluid of light has been studied in a nonlocal nonlinear liquid through the measurement of the dispersion relation of its elementary excitations[17] and the detection of a vortex nucleation in the wake of an obstacle[18]. The stimulated emission of dispersive shock waves in nonlinear optics was also studied in the context of light superfluidity[13]. However, one of the most striking manifestations of superfluidity—which is the ability of a fluid to move without friction[19]—has never been directly observed in a cavityless nonlinear-optics platform. A direct consequence of this feature is the absence of long-range radiation in a slow fluid flow past a localized obstacle. In optical terms, this corresponds to the absence of light diffraction from a local modification of the underlying refractive index in the plane transverse to the propagation. On the contrary, in the "frictional", nonsuperfluid regime, light becomes sensitive to such an index modification and diffracts while hitting it.

Here, we report a direct observation of a superfluid regime characterized by the absence of long-range radiation from the obstacle. This regime is usually associated with the cancellation of the drag force experienced by the obstacle, as studied for $^4$He[20], ultracold atomic gases[21–25], or cavity exciton-polaritons[26–29]. In our cavityless all-optical system, we extract on the one hand a quantity corresponding to the optical analog of this force and measure on the other hand the associated obstacle displacement. For the first time, at least within the framework of fluids of light, we observe that this displacement is nonzero in the nonsuperfluid case and tends to vanish while reaching the superfluid regime.

## Results

**Hydrodynamics of light**. We make use of a biased photo-refractive crystal which is, thanks to its controllable nonlinear optical response, convenient for probing the hydrodynamic behavior of light[13,30–32]. As sketched in Fig. 1a and detailed in

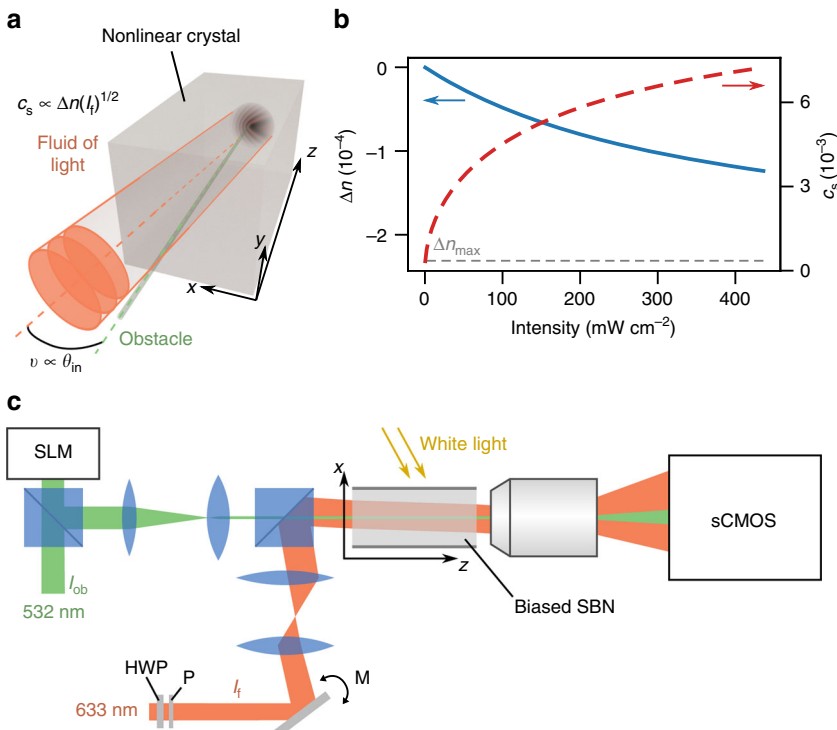

**Fig. 1** Realization of a fluid of light in a propagating geometry and nonlinear response of the bulk crystal. **a** Sketch of the fluid of light (red beam) flowing past an obstacle (green beam). The input velocity $v$ of the fluid of light is proportional to the input angle $\theta_{in}$. The sound velocity $c_s$ depends on the intensity $I_f$ of the red beam. **b** Blue curve. Calculated optical-index variation $\Delta n$ with respect to a laser intensity $I$ for the nonlinear photorefractive response of the medium. Red dashed curve. Corresponding sound velocity $c_s$. **c** Experimental setup. The green beam is shaped by the spatial light modulator (SLM) to create a $z$-invariant optical defect acting as a localized obstacle in the transverse plane. The red beam is a large gaussian beam and creates the fluid of light. $I_f$ is controlled by a half-waveplate (HWP) and a polarizer (P). $\theta_{in}$ is tuned by rotating a mirror (M) imaged at the input of the crystal via a telescope. Both are propagating simultaneously through a biased SBN photorefractive crystal and imaged on a sCMOS camera. The white light controls the saturation intensity of the crystal

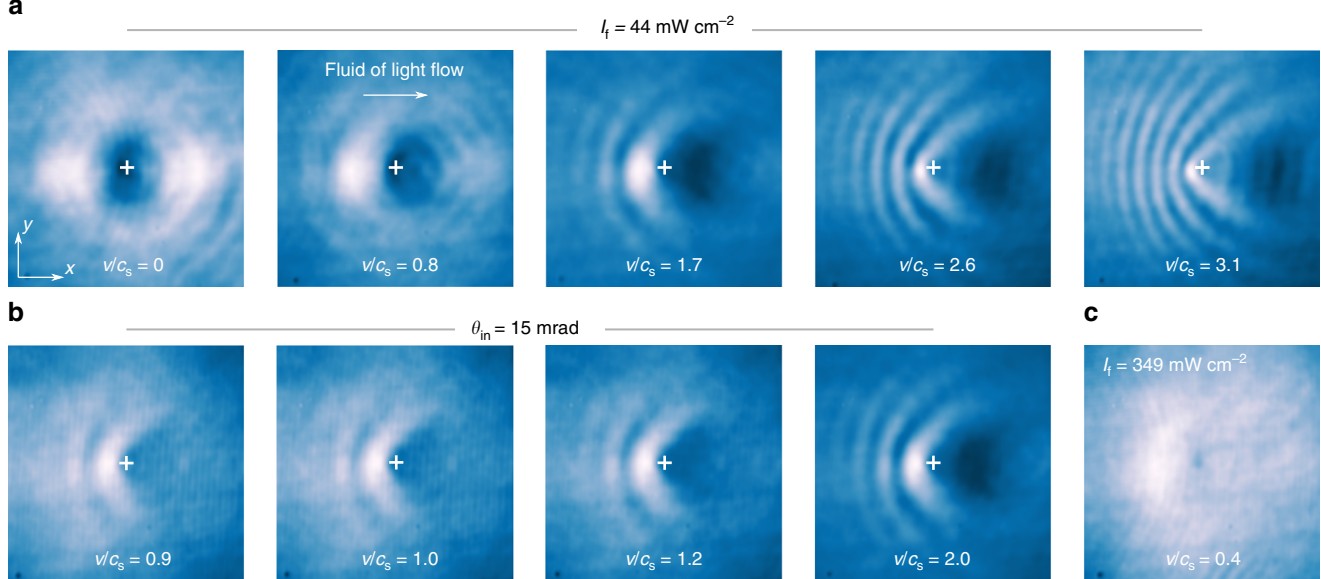

**Fig. 2** Spatial distribution of the output intensity of the fluid of light for various input conditions. The fluid of light flows from left to right. The white crosses at the center of the images indicate the position of the obstacle. Each image is $330 \times 330\ \mu m^2$. **a** At a fixed input intensity $I_f$, the input angle $\theta_{in}$ of the beam creating the fluid of light is tuned to vary the Mach number $v/c_s$ from 0 to 3.1. **b** Similarly, at a fixed input angle $\theta_{in}$, $I_f$ is progressively decreased to change $v/c_s$ from 0.9 to 2.0. **c** For large $I_f$, the fluid of light is clearly in the superfluid regime at $v/c_s = 0.4$. The remaining lack of uniformity upstream from the obstacle is attributed to propagation losses due to linear absorption

Fig. 1c, a local drop of the optical index is photo-induced by a narrow beam in the crystal and creates the obstacle. Simultaneously, a second, larger monochromatic beam is sent into the crystal and creates the fluid of light. The propagation of the fluid-of-light beam in the paraxial approximation is ruled by a two-dimensional Gross-Pitaevskii-type equation (also known as a nonlinear Schrödinger-type equation):

$$i\partial_z E_f = -\frac{1}{2n_e k_f}\nabla^2 E_f - k_f \Delta n(I_{ob})E_f - k_f \Delta n(I_f)E_f \qquad (1)$$

The propagation coordinate $z$ plays the role of time. The transverse-plane coordinates $\mathbf{r} = (x, y)$ span the two-dimensional space in which the fluid of light evolves. The propagation constant $n_e k_f = n_e \times 2\pi/\lambda_f$ of the fluid-of-light beam propagating in the crystal of refractive index $n_e$ is equivalent to a mass; the associated Laplacian term describes light diffraction in the transverse plane. The density of the fluid is given by the intensity $I_f \propto |E_f|^2$. Its velocity corresponds to the gradient of the phase of the optical field. At the input, it is simply given by $v \simeq \theta_{in}/n_e$, with $\theta_{in}$ the angle between the fluid-of-light beam and the $z$ axis (see Supplementary Note 1 for more details). The local refractive index depletion $\Delta n[I_{ob}(\mathbf{r})] < 0$ is induced by the obstacle beam of intensity $I_{ob}(\mathbf{r})$. The self-defocusing nonlinear contribution $\Delta n(I_f) < 0$ to the total refractive index provides repulsive photon–photon interactions and ensures robustness against modulational instabilities[33]. From the latter, we define an analog healing length $\xi = [n_e k_f \times k_f |\Delta n(I_f)|]^{-1/2}$, which corresponds to the smallest length scale for intensity modulations, and an analog sound velocity $c_s = (n_e k_f \times \xi)^{-1} = [|\Delta n(I_f)|/n_e]^{1/2}$ for the fluid of light[4,16] (see Supplementary Note 1). The photorefractive nonlinear response of the material, $\Delta n(I)$, is plotted in blue in Fig. 1b as a function of the laser intensity $I$ (see the Methods section for details). In the same figure, the red dashed curve represents the speed of sound $c_s(I)$.

When the obstacle is infinitely weakly perturbing, Landau's criterion for superfluidity[19] applies and the so-called Mach number $v/c_s$ mediates the transition around $v/c_s = 1$ from a

nonsuperfluid regime at large $v/c_s$ to a superfluid regime at low $v/c_s$. Generally this condition is not fulfilled and the actual critical velocity is lower than the sound velocity $c_s$[4,34]. This is the case in the present work for two main reasons. First, we consider a weakly but finite perturbing obstacle. It means a small variation of the refractive index $\Delta n[I_{ob}(\mathbf{r})] = -2.2 \times 10^{-4}$ and a radius of 6 μm comparable to $\xi$ (see Methods section and Supplementary Note 2). Note however that the perturbation is weak enough for the transition not to be blurred by the emission of nonlinear excitations like vortices or solitons. Second, remaining within Landau's picture, the speed of sound is here defined for $I_f$ measured at its maximum value, at $z = 0$, whereas the latter naturally suffers from linear absorption and self-defocusing along the $z$ axis.

**Probing the transition to superfluidity.** The ratio $v/c_s$ is controlled in the experiment both by the incidence angle $\theta_{in}$ and the input intensity $I_f$ of the fluid-of-light beam. Figure 2 presents typical experimental results for the spatial distribution of the light intensity observed at the output of the crystal for various input conditions. Figure 2a displays the output spatial distributions of intensity for different fluid velocities $v$ at a fixed speed of sound, $c_s = 3.2 \times 10^{-3}$. This allows to vary $v/c_s$ from 0 to 3.1. As $v$ increases, diffraction appears in the transverse plane, and progressively manifests as a characteristic cone of fringes upstream from the obstacle[14,16,35]. Another way to probe the transition is to fix the transverse velocity $v$ and to vary the sound velocity $c_s$ by changing the intensity of the fluid-of-light beam. Although the two ways of varying $v/c_s$ are not equivalent, as we shall discuss later, the results shown in Fig. 2b are similar with the interference pattern becoming more and more pronounced as $v/c_s$ increases. Figure 2c represents the intensity distribution at the output of the crystal for $v/c_s = 0.4$. Long-range radiation upstream from the obstacle is no longer present in this case, indicating a superfluid motion of light. The lack of uniformity of the intensity upstream from the obstacle is due to the intrinsic linear absorption of the material[29].

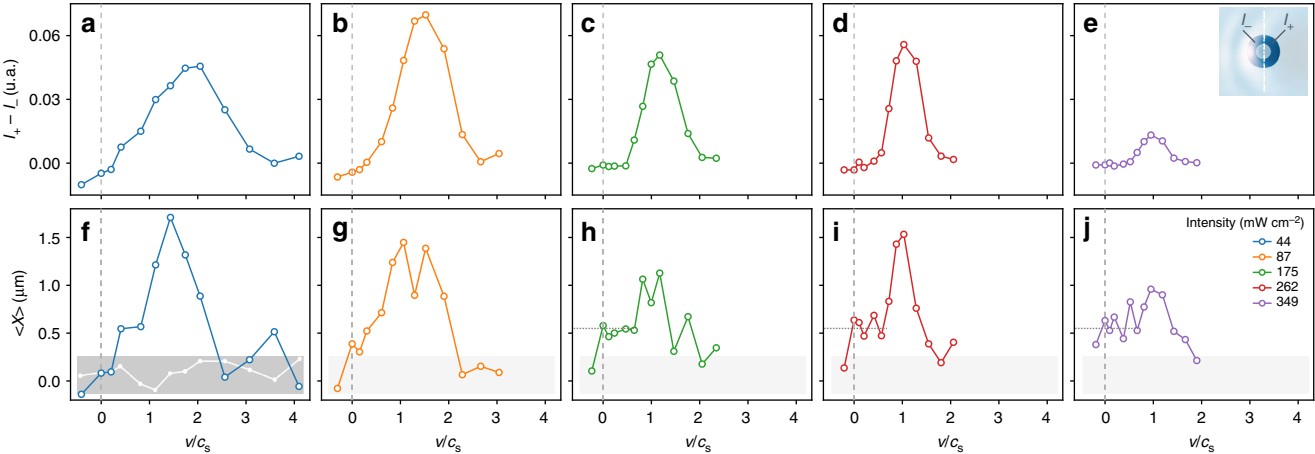

**Fig. 3** Optical analog of the drag force exerted by the fluid and associated displacement of the obstacle. **a–e** Local intensity difference $I_+ - I_-$ extracted from the experimental images of the intensity of the fluid-of-light beam measured at the crystal's output for various input conditions ($I_f$ ranging from 44 (**a**) to 349 (**e**) mW cm$^{-2}$ and $v/c_s$ ranging from −0.41 to 4.10). Inset of **e**: the original image is cropped around the optical defect and integrated over two regions, downstream ($I_-$) and upstream ($I_+$). The typical integration area is of the order of $\xi$. The grey dotted line corresponds to $v/c_s = 0$. **f–j** Measurement of the transverse displacement of the obstacle induced by the local modulation of the intensity of the fluid of light for the same input conditions as for figures **a–g**. Grey boxes define the typical uncertainty in the measured quantities, the white points in **f** corresponding to the displacement along the $y$ axis for $I_f = 44$ mW cm$^{-2}$, which is expected to be zero

**Drag-force and obstacle displacement**. In the supersonic regime, the intensity modulation of the fluid of light flowing around the obstacle induces a local optical-index modification of the material. This modification influences the propagation of the beam responsible for the obstacle, for which a transverse displacement is expected. On the contrary, in the superfluid regime, the absence of long-range intensity perturbations implies no local variation of the optical index and then one does not await for any displacement of the obstacle beam.

As theoretically investigated in[36] for a material obstacle (here, we rather consider an all-optical obstacle), the local intensity difference for the fluid of light between the front ($I_+$) and the back ($I_-$) of the obstacle, $I_+ - I_-$, is proportional to the dielectric force experienced by the obstacle. This force turns out to be closely analogous to the drag force that an atomic Bose-Einstein condensate exerts onto some obstacle. Figure 3a–e depicts the variation of $I_+ - I_-$, measured at the output of the crystal, as a function of $v/c_s$ for various initial conditions. As illustrated in the inset of Fig. 3e, both intensities are integrated over a typical distance of the order of $\xi$ surrounding the obstacle. For all intensities, we observe a rather smooth, but net transition for $v$ slightly smaller than $c_s$. The increasing tendency for low Mach numbers is associated to linear absorption, as discussed in the context of cavity quantum fluids of light[26,27,29]. The well-known decreasing tendency at large Mach numbers is also observed. Indeed, the obstacle can always be treated as a perturbation at large velocities and the associated drag force resultingly decreases[37]. As the intensities increase, one can see that the local intensity difference sticks to zero for non-zero values of $v/c_s$, as predicted for the drag fore in a superfluid regime. Moreover, Fig. 3a–e shows that the curves with different intensities $I_f$, although renormalized by the respective sound velocity $c_s$, do not fall on a single universal curve. This is due to the fact that changing the intensity also affects crucial quantities like the healing length $\xi$ and the relative strength of the obstacle with respect to the nonlinear term, $\Delta n(I_{ob})/\Delta n(I_f)$. While the drop of this force is among the main signatures of superfluidity in material fluids, so far this is the first experiment on fluids of light investigating it.

To go one step further, we probe the corresponding transverse displacement of the obstacle, independently on the measurement of $I_+ - I_-$. By assuming that the transverse component of the fluid-of-light beam is non-zero only along the $x$ axis, we denote by $\langle x \rangle = \int x |E_{ob}|^2 dx$ the position of the centroid of the obstacle beam. Using an optical equivalent of the Ehrenfest relations, one can derive the following equation of motion (see Supplementary Note 3 for full derivation):

$$n_e \partial_{zz} \langle x \rangle = \partial_x [\Delta n(I_f)]. \tag{2}$$

This means that the all-optical obstacle is sensitive to the surrounding refractive index potential that mainly results from the spatial distribution of intensity of the beam creating the fluid of light. It thus might move of a distance $d = \langle x \rangle - x_0$ from its initial position $x_0$ in the transverse plane. The measurement of $d$ for various conditions in the case of an obstacle evolving in a fluid of light at rest allows to validate such an experimental approach and to extract experimental parameters as $I_{sat}$ and $\Delta n_{max}$ (see Methods section and Supplementary Note 3).

Figure 3f–j shows the transverse displacement measured in a moving fluid of light varying the Mach number $v/c_s$ for different initial conditions. To take into account the gaussian shape of $I_f$, we subtract, for each data point, the displacement measured when the influence of the obstacle on the fluid of light is negligible (i.e., very low $I_{ob}$), as illustrated in Supplementary Note 3. The white points in Fig. 3f correspond to the displacement along the $y$ axis and is expected to be zero. The grey boxes thus define the typical uncertainty in the measured quantities. The fluctuation are attributed to the inherent imperfections of the fluid-of-light beam. We observe that the transverse displacement of the obstacle behaves very similarly to the intensity difference $I_+ - I_-$ displayed in Fig. 3a–e. That is, an increasing displacement from almost zero in the deeply subsonic regime to maximum signal, and then a decreasing tendency in the supersonic regime. We also measured an opposite transverse displacement for negative $v/c_s$. Note that in this case, due to cavity effects, large interference patterns blurred the signal and did not allow to perform quantitative analysis (see Supplementary Note 4). The fact that the displacement is not purely zero in the superfluid regime is likely due to the displacement acquired during the non-stationary regime at early stage of the propagation (see Supplementary Note 5 for qualitative discussion supported by numerical

simulations). This is, to the best of our knowledge, the first observation of the displacement of an all-optical obstacle in a fluid of light.

## Discussion

We reported a direct experimental observation of the transition from a "frictional" to a superfluid regime in a cavityless all-optical propagating geometry. We performed a quantitative study by extracting an optical equivalent of the drag force that the fluid of light exerts on the obstacle. This result is in very good agreement with an independent measurement that consists in studying the transverse displacement of the obstacle surrounded by the fluid of light. We restricted the present study to the case of a weakly perturbing obstacle but our experimental setup allows to reach the turbulent regime associated to vortex generation through the induction of a greater optical-index depletion. On the other hand, a different shaping of the beam creating the obstacle will allow to generate any kind of optical potential and to extend the study to imaging through disordered environments.

## Methods

**Experimental setup.** The nonlinear medium consists in a $5 \times 5 \times 10$ mm$^3$ strontium barium niobate (SBN:61) photorefractive crystal additionally doped with cerium (0.01%) to enhance its photoconductivity[38] albeit it induces linear absorption (3.2 dB/cm). The basic mechanism of the photorefractive effect remains in the photogeneration and displacement of mobile charge carriers driven by an external electric field $E_0$. The induced permanent space-charge electric field thus implies a modulation of the refractive index of the crystal[39], $\Delta n(I, \mathbf{r}) = -0.5 n_e^3 r_{33} E_0 / [1 + I(\mathbf{r})/I_{sat}]$, where $n_e$ is the optical refractive index and $r_{33}$ the electro-optic coefficient of the material along the extraordinary axis, $I(\mathbf{r})$ is the intensity of the optical beam in the transverse plane $\mathbf{r}(x, y)$, and $I_{sat}$ is the saturation intensity which can be adjusted with a white light illumination of the crystal. The blue curve in Fig. 1b shows the saturable nonlinear response of the material $\Delta n(I)$ against the laser intensity $I$. The red dashed curve represents the sound velocity $c_s(I)$ for the saturable nonlinear response of the material $\Delta n(I)$. The maximum value of the optical index variation is theoretically $\Delta n_{max} = -2.32 \times 10^{-4}$ for $E_0 = 1.5$ kV cm$^{-1}$.

**Shaping the fluid of light and obstacle beams.** Making use of a spatial light modulator, we produce a diffraction-free Bessel beam ($\lambda_{ob} = 532$ nm, $I_{ob} = 7.6$ W cm$^{-2} \gg I_{sat}$, green path in Fig. 1c). The latter creates the obstacle with a radius of 6 μm (comparable to $\xi = 6.2$ μm obtained for $I_f = 349$ mW cm$^{-2}$) that is constant all along the crystal and aligned with the $z$-direction. From Fig. 1b, the propagation of the obstacle beam into the crystal induces a local drop $\Delta n(I_{ob}) = -2.2 \times 10^{-4}$ in the refractive index. A second laser ($\lambda_f = 633$ nm, red path in Fig. 1c) delivers a gaussian beam whose radius is extended to 270 μm and which corresponds to the fluid-of-light beam. Both laser beams are linearly-polarized along the extraordinary axis to maximize the photorefractive effect. We vary the flow velocity $v$ by changing the input angle $\theta_{in}$ of the fluid-of-light beam with respect to the propagation axis $z$ (see Fig. 1a). The accessible range, tuned by rotating a mirror imaged at the input of the crystal via a telescope, goes from $\theta_{in} = 0$ to $\pm 23$ mrad, corresponding to $v$ ranging from $v = 0$ to $v = \pm 1.3 \times 10^{-2}$. The sound velocity $c_s$ is controlled by the input intensity of the beam which can be tuned from $I_f = 0$ to 350 mW cm$^{-2}$ via a half-waveplate and a polarizer. The maximum value for $c_s$ is $6.8 \times 10^{-3}$, as plotted in Fig. 1b. For the detection part, a ×20 microscope objective and a sCMOS camera allow to get the spatial distribution of the near-field intensity of the beams at the output of the crystal.

**Displacement of the obstacle beam in the fluid of light beam at rest.** In order to validate our experimental approach, we consider the linear propagation of the green beam creating the obstacle in the optical potential $\Delta n(I_f)$ photo-induced by the fluid-of-light beam at rest ($\theta_{in} = 0$). In the paraxial approximation, the propagation equation reads

$$i \partial_z E_{ob} = -\frac{1}{2 n_e k_{ob}} \nabla^2 E_{ob} - k_{ob} \Delta n(I_f) E_{ob}, \quad (3)$$

with notations similar to the ones used in Eq. (1). By assuming that the transverse component of the fluid-of-light beam is non-zero only along the $x$ axis, we denote by $\langle x \rangle = \int x |E_{ob}|^2 dx$ the position of the centroid of the obstacle beam. Using an optical equivalent of the Ehrenfest relations (see Supplementary Note 3 for full derivation), one can derive from Eq. (3) the following equation of motion: $(n_e k_{ob}) \partial_{zz} \langle x \rangle = -\partial_x [-k_{ob} \Delta n(I_f)]$. Assuming that $\Delta n$ is $z$-independent, which is valid in the here-considered linear propagation of the obstacle beam, we readily

obtain

$$d = \langle x(z) \rangle - x_0 = \frac{1}{2} [\partial_x \Delta n(I_f)/n_e] z^2 \quad (4)$$

where $x_0$ is the initial position of the obstacle. This displacement is interpreted as the consequence of a force deriving from the optical potential $-k_{ob} \Delta n(I_f)$, and acting on the obstacle.

The experimental measurement of $d$, for various intensities $I_f$ and positions $x_0$, is presented in Supplementary Fig. 2. The experimental data are fitted, using the above expression, the saturation intensity and the maximum refractive index modification being the fitting parameters. We extract $I_{sat} = 380 \pm 50$ mW cm$^{-2}$ and $\Delta n_{max} = 2.5 \pm 0.4 \times 10^{-4}$. It is worth mentioning that the value of $I_{sat}$ is used for the calculation of $\Delta n(I)$ and its deriving quantities (i.e., $c_s$ and $\xi$).

**Data availability**. The data supporting the findings of this study are available within the article and the associated Supplementary Information. Any other data is available from the corresponding author upon request.

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

## Acknowledgements

The authors acknowledge helpful contributions from M. Garsi during the early stage of this work. We also thank I. Carusotto, V. Doya, F. Mortessagne, N. Pavloff, and P. Vignolo for helpful discussions. M.A. is grateful to P. Leboeuf who was very enthusiastic about the idea of superfluid motion of light. This work has been supported by the the Region PACA and the French government, through the UCA^JEDI Investments in the Future project managed by the National Research Agency (ANR) with the reference number ANR-15-IDEX-01. P.-É.L. was funded by the Centre National de la Recherche Scientifique (CNRS), the ANR under Grant No. ANR-14-CE26-0032 LOVE, and the Universit?é de Cergy-Pontoise.

## Author contributions

C.M., O.B., and M.B. performed the experiments and analyzed the data. C.M., M.A., P.É.L., and M.B. developed the theory. All authors participated in the discussions and in writing the paper.

## Additional information

**Competing interests:** The authors declare no competing interests.

