## [Peer Review File · Nature Communications]

Reviewers' comments:

Reviewer #1 (Remarks to the Author):

This is the first example I have seen of a direct measurement of the drag force in the non-superfluid regime of a photon fluid. The idea of using an optical obstacle that is in turn influenced by the presence or absence of super flow is highly novel approach and the results will influence other in the field and beyond (BEC, cavity photon fluids etc.)

I found the paper to be well written and self-contained. The measurements were convincing and not obscured by noise, and the theoretical analysis was clear. My only minor gripe is why there is no self-phase modulation term $\Delta n(I_{ob})$ included in Eq. 3 of the methods section, though I don't think this will effect the main result in Eq. 4. There are also some small places where the English could be improved and the authors should check this.

In summary I recommend publication of this paper in Nature Communications after the authors have considered the minor issues above.

Reviewer #2 (Remarks to the Author):

Authors report an all-optical analogy of superfluid flowing around an obstacle based on Nonlinear Schrödinger equation type system. Particularly they performed a detailed studies on the dragging force on the obstacle based on its displacement. The interesting points lay on the existence of drag force before/after the superfluid point.

In terms of novelty, the current work is definitely not the first one to conduct such studies, see optical superfluid around an obstacle in 1D - PRL 104, 073903, 2D version - Frontiers in Optics, FThD4,2008. They both studied the same problem, however, the current work focuses on the drag force associated to the superfluid flowing around the obstacle, which is new & interesting.

However, in terms of experiment, I think the current observations maybe weak to justify authors' claim. For starters, Fig. 3 is so messy & blur to tell the displacement of the obstacle. Plus, technically I don't think $I_+ - I_-$ is a good term for characterizing the displacement. There're some messy distortion around the defect cylinder due to photorefractive effect, which both green pump & red probe can contribute to. Some other terms may be considered, for example, the fringes' period before, top/ bottom, behind (above the wake regime), because they experiences different forces. In this sense, 1D experiments may be even beneficial to be shown & compared, cause there's no flowing-around action.

Overall, the work is interesting, but it needs be revised to address the above points in order to be published.

Reviewer #3 (Remarks to the Author):

In their manuscript, Claire Michel and coworkers present experimental results about the propagation of a fluid of light in a cavity-less geometry. They consider a paraxial geometry where the z-direction play the role of time. The Kerr nonlinearity of the crystal is responsible for the photon-photon interactions. In this geometry, the authors are able to study the transverse propagation of the light fluid in presence of an optically-generated defect. In this experimental set-up, the authors have observed a phenomenology similar to what obtained in planar microcavity

systems. The original study of the present work is the measurement of the the drag force using two different methods. One method is based on the local intensity difference around the obstacle, while the second one is a direct measurement of the defect displacement.

I believe that the paper presents an interesting approach. Moreover the description of the methods is very detailed.

However, I am puzzled by the main results, which are shown in Fig. 3. in order to be convinced enough by the data, the authors need to carefully address the comments below:

i) The authors claim that the two methods to extract the drag force agree. However, from Fig 3.b I can see a finite displacement d for small values of v/c_s . Instead, in Fig. 3.a. the curve is relatively much closer to zero.

ii) The discussion of the negative velocity is confusing. Why are the authors not showing a larger interval of negative values for the normalized velocity ? Are there artifact creating an unwanted asymmetry ?

iii) In the Supplementary material, there are a few theoretical curves. Why have the authors not shown the results of the theory for Fig. 3 ? Do they agree with the experiments or not ?

Point-to-point reply to the reviewers

We thank all three reviewers for their constructive remarks about our manuscript. We are please to provide in this document a detailed point-to-point reply to their comments. The associated modifications in the manuscript are highlighted in blue, for the sake of clarity, except for two references as the bibliography is generated automatically.

ANSWER TO REVIEWER 1

We would like to thank Reviewer 1 for his/her very positive feedback especially concerning the originality and the broad scope of our work. He/her moreover notices on the one hand, that the measurements are convincing and not obscured by noise, and on the other hand, that the theoretical analysis is clearly exposed. He/her still has some minor remarks that we are pleased to address in details in the following.

Comment 1. “My only minor gripe is why there is no self-phase modulation term $\Delta n(I_{\text{ob}})$ included in Eq. 3 of the methods section, though I don’t think this will effect the main result in Eq. 4.”

As mentioned in the manuscript, the Eq. 3 considered in the Methods section describes a special case where the obstacle propagates *linearly* in a potential induced by the fluid of light at rest (typically a gaussian shape). Hence, we consider an intensity I_{ob} low enough so that any nonlinear effect can be neglected, including the self-phase modulation effect. Consequently, we keep unchanged the paragraph “Shaping the fluid of light and obstacle beams” of the Methods section. The aim of these measurements is to show that the obstacle beam is indeed sensitive to the environment induced by the fluid-of-light beam and might move in the transverse direction with respect to its initial position. It allows i) to validate our experimental approach regarding the obstacle displacement and ii) to extract typical experimental parameters (I_{sat} and Δn_{max}). To clarify this approach, we provide an additional explanation in the paragraph “Obstacle displacement in a moving fluid of light” in Section S.3 of the Supplementary.

Nevertheless, we fully agree with Reviewer 1 that in the general case the obstacle beam is also propagating in the nonlinear regime and that the nonlinear term $\Delta n(I_{\text{ob}})$ should be taken into consideration. More precisely, the dynamics of our system is described by two coupled nonlinear Schrödinger equations as follows :

$$\begin{aligned} i\partial_z E_f &= -\frac{1}{2n_e k_f} \nabla^2 E_f - k_f \Delta n(I_{\text{ob}}) E_f - k_f \Delta n(I_f) E_f \\ i\partial_z E_{\text{ob}} &= -\frac{1}{2n_e k_{\text{ob}}} \nabla^2 E_{\text{ob}} - k_{\text{ob}} \Delta n(I_f) E_{\text{ob}} - k_{\text{ob}} \Delta n(I_{\text{ob}}) E_{\text{ob}} \end{aligned}$$

Thus, for the sake of clarity, we completed the theoretical description by considering the general case as suggested by Reviewer 1. More precisely, eq. 2 of the main text remains the same, but its full derivation is detailed in the general case within eqs. S.3 – S.7 in the Supplementary information. The paragraph concerning the linear case has also been modified consequently. It is worth mentioning that the influence of the Self-Phase Modulation on the obstacle beam shape is actually negligible with respect to the displacement induced by the fluid of light.

Comment 2. “There are also some small places where the English could be improved and the authors should check this.”

We thank Reviewer 1 for pointing out this issue regarding the English. We read again our manuscript and checked it. Some sentences have been rephrased. We hope that this will meet the standard criteria.

ANSWER TO REVIEWER 2

We would like to thank Reviewer 2 for noticing that our work represents a new and interesting study of light superfluidity through the measurement of the drag force experienced by an obstacle. We also would like to thank him/her for his/her constructive remarks and concerns about several points of our study. We are pleased to provide a list of answers to his/her criticisms below.

Comment 1. "In terms of novelty, the current work is definitely not the first one to conduct such studies, see optical superfluid around an obstacle in 1D - PRL 104, 073903, 2D version - Frontiers in Optics, FThD4,2008. They both studied the same problem, however, the current work focuses on the drag force associated to the superfluid flowing around the obstacle, which is new and interesting."

We agree with Reviewer 2 that our work is not the first one to consider the motion of a quantum fluid of light through an obstacle. We actually knew these important studies and regret not to have cited them properly in the manuscript. We thank Reviewer 2 for noticing this mistake that we have corrected in the current version (refs. [30] and [31]). Note however that, in these previous works, the different regimes of flow were not explicitly identified. In particular, no critical velocity was identified and the superfluid regime was not really reached. Indeed, the mentioned work in 1D focuses on the tunneling of a wave through an obstacle and if the superfluid regime were reached, the transmission coefficient would have been measured to be unity. Our contribution goes a way further since we observe and identify the different transport regimes qualitatively and quantitatively (measurement of the drag force imposed by the fluid to an obstacle and its corresponding displacement).

Comment 2. "However, in terms of experiment, I think the current observations maybe weak to justify authors' claim. For starters, Fig. 3 is so messy and blur to tell the displacement of the obstacle."

Referee 2 is right, the way of presenting our results on figure 3 was probably not appropriate. For the sake of clarity, we have now plotted the results in separated plots, each of them corresponding to a given intensity of the fluid of light beam. This makes the analysis much easier for the reader since the displacement appears more clearly. We thank the referee for pointing out this weakness that we hope to have fixed now.

We would like to remind Reviewer 2 that the white points in figure 3.f represent the displacement of the obstacle in the y direction, along which it is not supposed to move. It fixes the typical error that one can do on a given measurement along the x direction, and is highlighted by the grey box on the other graphs. As one can see, the data points are clearly out of the grey boxes, and then correspond to an actual displacement, and not some noisy measurement. Moreover, the direct comparison with the optical drag force is even clearer with this representation, as one sees at first glance the correspondance between the two quantities, i) the drag force experienced by the obstacle and ii) the associated displacement of the obstacle. It is important to note that in the superfluid regime, there is still a plateau corresponding to a residual displacement which is clearly out of the noise. We explain this plateau in section S.4 of the Supplementary information, and emphasize it in the corrected figure 3, by a black dotted-line, to make it even clearer.

Comment 3. "Plus, technically I don't think $I_+ - I_-$ is a good term for characterizing the displacement. There're some messy distortion around the defect cylinder due to photorefractive effect, which both green pump and red probe can contribute to. Some other terms may be considered, for example, the fringes' period before, top/ bottom, behind (above the wake regime) , because they experiences different forces. In this sense, 1D experiments may be even beneficial to be shown and compared, cause there's no flowing-around action."

We understand Reviewer 2' scepticism about the way we quantify the drag force. It may appear inaccurate to associate the drag force to a local quantity such as the difference of intensity upstream and downstream of the obstacle, especially when experimental noise is present. We would then like to answer this important remark in three points:

1. Although not really mentioned in the main part of the manuscript, this procedure is very standard and was widely used in both communities of quantum fluids of matter and light to extract the drag force from density or intensity maps (see for instance [Frisch, T., Pomeau, Y. and Rica, S., Phys. Rev. Lett. 69, 11 (1992)], [Pavloff, N., Phys. Rev. A 66, 013610 (2002)], [Larré, P.-É. and Carusotto, I., Phys. Rev. A 91, 053809 (2015)]). From the quantum mechanical perspective, this basically comes from the fact that the average value of the force operator experienced by the fluid due to the obstacle is given by (in 1D for the sake of simplicity but the generalization is trivial) $F = \langle \psi | -dV/dx | \psi \rangle = \int dx \psi^*(x) (-dV(x)/dx) \psi(x) = - \int dx |\psi(x)|^2 dV(x)/dx$,

where $V(x)$ is the potential representing the obstacle and $\rho(x) = |\psi(x)|^2$ is the density associated to the wave function $\psi(x)$. A simple integration by part allows one to understand the origin of the difference of densities as a signature of a drag force. The reasoning presented here elaborates on the similarities shared by the NLSE of nonlinear dielectric media and the GPE of dilute atomic Bose gases. It suggests that the intensity difference $I_+ - I_-$ is the suitable observable to probe in order to get insights into what we call the drag force. This is in fact fully confirmed by a microscopic derivation of the total dielectric force [S. M. Barnett and R. Loudon, *J. Phys. B: At. Mol. Opt. Phys.* 39, S671 (2006); R. Loudon and S. M. Barnett, *Opt. Express* 14, 11855 (2006)] experienced by the refractive-index depletion we generate in the crystal. After time averaging its instantaneous value over a few optical periods of the carrier (what our detectors practically do), this force becomes proportional to the integral of the optical intensity times the gradient of our refractive-index depletion, which is nothing but the drag force of quantum fluids of matter after replacing the intensity by the density, and the refractive-index depletion by the potential of the obstacle [P.-É. Larré and I. Carusotto, *Phys. Rev. A* 91, 053809 (2015)]. With this, we definitely have no doubt that $I_+ - I_-$ is what we have to measure in the experiment, difficult though that might be in our 2D setup.

2. Our experimental intensity maps are actually not so messy. As explained in the text, the intensity is integrated in the vicinity of the obstacle and is not sensitive to long distance radiations in the supersonic regime for instance. We have checked that our results are robust when slightly modifying the radius of integration which of course has to be less than or comparable to the healing length of the quantum fluid of light. This ensures that this quantity is insensitive to spurious perturbations since the healing length is the physical spatial resolution of the quantum fluid.
3. Finally, we corroborate three different observations to characterize the different regimes of transport. First a clear qualitative distinction is observable on the intensity plots on Fig. 2 of the main text. Depending on the Mach number one clearly sees the difference between the superfluid and non-superfluid regimes. In the later case, long distance radiation is clearly present and is a signature of a drag force (whenever the intensity profile is asymmetrical there is a drag force). Then, we extract from these figures, a quantity which is proportional to the drag force, namely the difference of intensities. But not only this quantity is in agreement with the qualitative picture, it allows us to extract the good order of magnitude of the critical velocity and it is very well correlated to the actual displacement of the obstacle through the influence of the drag force. This is the consistence of all these results that makes us believe our measurement is trustworthy.

ANSWER TO REVIEWER 3

Reviewer 3 recognizes the originality of our present work through the measure of the optical drag force by means of two independent and uncorrelated methods and we thank him/her for that comment. He/She raises some points that we are pleased to address here after.

Comment 1. “The authors claim that the two methods to extract the drag force agree. However, from Fig 3.b I can see a finite displacement d for small values of v/c_s . Instead, in Fig. 3.a. the curve is relatively much closer to zero.”

Indeed, this is exactly the aim of the section S.4 of the supplementary information entitled “Qualitative discussion on the obstacle extra displacement”. As the question is asked, it certainly means that the explanation is not clear enough. Thus, we highlight here the main arguments.

In Fig. 3b of the main text, we observe that the displacement is not purely zero in the superfluid regime for large intensities. There is still a plateau indicating a residual displacement (which has been emphasized by a black dotted-line in the new figure 3). It comes from a preliminary interaction at the very beginning of the beams propagation, before reaching the stationary regime. In order to justify this statement, we performed numerical simulations of the propagation of the beams in the crystal. They are presented in Fig. S.4b of the Supplementary. It is clear that the system presents an asymmetry of the spatial distribution of intensity. This asymmetry implies a non-zero drag force resulting in a displacement of the obstacle. From 7mm, the system enters in a superfluid regime. The intensity distribution is then uniform and no force any longer applies on the obstacle. Still, the previously accumulated displacement remains, and is observed in Fig. 3 of the main text.

Comment 2. “Why are the authors not showing a larger interval of negative values for the normalized velocity ? Are there artifact creating an unwanted asymmetry ?”

Reviewer 2 is right, there are some measurement artifacts which do not allow any serious quantitative analysis for negative velocities. We provide a detailed reply about the origin of these artifacts in the following.

To avoid typical cavity effects (large interferences appearing when the light propagates perpendicularly to the faces of the sample), the crystal was slightly tilted with an angle θ_0 with respect to the propagation direction. This situation is illustrated in Fig. A1. The positive values of the fluid of light velocity are defined for input angles $\theta_{in} > 0$ (Fig. A1, top). In this case, no interference effects are expected. However, for negative velocities defined for $\theta_{in} < 0$ (Fig. A1, bottom), as θ_{in} approaches θ_0 , large interferences pattern appear. Typical images of the output intensity, shown in Fig A2.a, exhibit such patterns (pointed out by the red arrows). Note that this set of data can not be compared directly with the figure 2 of the manuscript since the experimental conditions are different (mainly, the acquisitions were performed with a 4x imaging microscope objective instead of a 20x in the manuscript). Qualitatively, the images look very similar with the disappearance of the diffraction pattern as v/c_s tends to zero. The associated obstacle

FIG. A1. **Experimental configurations.** To avoid cavity effects, the crystal is slightly tilted with an angle θ_0 . For positive fluid of light velocities (top), no interferences patterns are expected. For negative velocities (bottom), as θ_{in} approaches θ_0 , large interferences pattern appear.

FIG. A2. **Illustration of the cavity effects appearing at negative v/c_s .** (a) Spatial distribution of the output intensity of the fluid of light for various negative v/c_s at $I = 262 \text{ mW.cm}^{-2}$. The red arrows denote large scale interferences. (b) Measurement of the transverse displacement of the obstacle induced by the local modulation of the intensity of the fluid of light for various input conditions. (c) Measurement of the transverse displacement of the obstacle along the y axis, which is expected to be zero. This defines the typical uncertainty in the measured displacements.

displacements, illustrated in Fig. A2.b, also present similarities with an increase (resp. decrease) at low (resp. high) Mach number and comparable values (about $1.5 \mu\text{m}$ maximum). However, as shown in Fig. A2.b which represents the displacement along the y axis expected to be zero, the measurement's uncertainty is much larger (at least twice) than for the results presented in the manuscript. In these conditions, the cavity effects blur the signal and forbid any serious quantitative analysis. It thus becomes tough to extract sensitive quantities such as the transition threshold and the plateau. This is the reason why we focused our study on positive values of the fluid velocity only. We presented the results for a very small interval of negative velocities (actually not blurred by the large interferences) to show the tendency for the obstacle to move towards the opposite direction.

Note that these blurring cavity effects are inherent to any propagating geometries. One solution would have been to cut and polish, at a small angle, one face of the crystal.

We thank Reviewer 2 for pointing out this issue and hope that the additional information provided will convince him/her. We take his/her remark into consideration in the manuscript by changing the sentence “We also measured an opposite transverse displacement for negative v/c_s ” by “We also measured an opposite transverse displacement for negative v/c_s . Note that in this case, due to cavity effects, large interference patterns blurred the signal and did not allow to perform quantitative analysis.”

Comment 3. “In the Supplementary material, there are a few theoretical curves. Why have the authors not shown the results of the theory for Fig. 3? Do they agree with the experiments or not?”

Reviewer 3 is right, we have indeed some analytical results for the displacement of the obstacle (paragraph “Obstacle displacement in the fluid of light at rest.” of the section S.3) but only when the obstacle does not affect the fluid of light propagation (i.e. when the obstacle beam propagates linearly, for a very low intensity I_{ob}). In this situation, the fluid of light beam is z -invariant, and the integration is possible from eq. S.8 to eq. S.9. As recalled in the reply to the Comment 1 of the Reviewer 1, the aim of these measurements is to show that the obstacle beam is indeed sensitive to the environment induced by the fluid-of-light beam and might move. It allows i) to validate our experimental approach regarding the obstacle displacement and ii) to extract typical experimental parameters (I_{sat} and Δn_{max}).

However, in the general case, the fluid of light is affected by the presence of the obstacle beam whose intensity

I_{ob} is not negligible anymore. In this situation, we do not have access to the analytical solution of eq. S.7 which corresponds to finding the solution of the equation $\partial_{zz}\langle x \rangle \propto \partial_x(\Delta n(x, y, z))$ where the spatial distribution of Δn is unknown along the propagation.

REVIEWERS' COMMENTS:

Reviewer #1 (Remarks to the Author):

After reading the new manuscript in combination with all of the reviewer comments, and the author's reply, I recommend that this paper be published in its present form.